# Over-activation of a nonessential bacterial protease DegP as an antibiotic strategy

Hyunjin Cho[1], Yuri Choi [1], Kyungjin Min [1], Jung Bae Son[1], Hyojin Park[1], Hyung Ho Lee[1] & Seokhee Kim [1]✉

Rising antibiotic resistance urgently begs for novel targets and strategies for antibiotic discovery. Here, we report that over-activation of the periplasmic DegP protease, a member of the highly conserved HtrA family, can be a viable strategy for antibiotic development. We demonstrate that tripodal peptidyl compounds that mimic DegP-activating lipoprotein variants allosterically activate DegP and inhibit the growth of an *Escherichia coli* strain with a permeable outer membrane in a DegP-dependent fashion. Interestingly, these compounds inhibit bacterial growth at a temperature at which DegP is not essential for cell viability, mainly by over-proteolysis of newly synthesized proteins. Co-crystal structures show that the peptidyl arms of the compounds bind to the substrate-binding sites of DegP. Overall, our results represent an intriguing example of killing bacteria by activating a non-essential enzyme, and thus expand the scope of antibiotic targets beyond the traditional essential proteins or pathways.

[1] Department of Chemistry, Seoul National University, 1 Gwanak-ro, Gwanak-gu, Seoul 08826, South Korea. ✉email: seokheekim@snu.ac.kr

Antibiotic resistance is one of the most urgent and growing public health threat. The level of resistance against effective antibiotics has been rising over the last several decades, whereas the number of new antibiotic drugs has dramatically decreased[1]. Furthermore, there has been limited innovation in antibiotic development: most antibiotics target only four major cellular pathways—maintenance of membrane, and biosynthesis of nucleic acids, of proteins, and of cell wall peptidoglycan—and newly developed antibiotics are mostly derivatives of old scaffolds, which are more susceptible to existing resistance mechanisms[2–4]. To overcome the current antibiotic crisis, it is of utmost importance to promptly develop new antibiotics with novel targets and mechanisms of action.

Targeting bacterial proteases provides unique opportunities in antibiotic discovery[5]. Protease inhibitors have been successfully applied to treat many human diseases, but antibiotics targeting proteases have not yet been used in clinic[6,7]. Typically, a protease targeting occurs through inhibition by an active-site inhibitor that mimicks native substrates. Distinctively, a class of antibiotics, acyldepsipeptides (ADEPs), kills bacteria by over-activating the cytoplasmic ClpP protease[8,9]. Particularly, in combination with other antibiotics, ADEPs kill persistent bacteria that are extremely difficult to eradicate with conventional antibiotics[10]. As such, although ADEPs have the potential to be used as alternatives to orthodox antibiotics, this novel approach of activating proteases has not been further explored, largely due to the lack of information on proper target proteases and activating molecules.

Here, we report data showing that the chemical over-activation of DegP, a highly conserved periplasmic heat-shock protease[11], can be an unprecedented strategy for development of new antibiotics. We reasoned DegP may be a good target for this approach for several reasons. First, DegP is an allosteric protease, whose activity is carefully regulated[12–15], and a hyperactive DegP variant kills bacteria[16]. Therefore, molecules that allosterically activate DegP may be exploited to trigger toxicity. Second, because DegP is an inherently promiscuous protease that degrades misfolded proteins, over-activated DegP may destroy various proteins that mediate essential processes in the bacterial envelope, such as cell wall maintenance, membrane construction, or cell division[17]. Third, although loss-of-function mutations in the *degP* gene can suppress the toxicity of activators, they may also reduce bacterial fitness against stress and pathogenesis. Indeed, DegP proteolysis is essential for cell viability at high temperatures that generate misfolded proteins[18,19], and the deletion of the homologous gene often reduces the virulence of pathogenic bacteria in animal models[20,21].

## Results and discussion

**Design of the DegP-activating molecules**. We previously reported that a variant of an outer membrane lipoprotein, Lpp^+Leu, which contains an additional leucine at the C-terminus, can function as an allosteric activator of wild-type DegP[22]. Because Lpp is a trimeric protein[23] and only the C-terminal region of Lpp^+Leu is critical for activity modulation[16,22], we reasoned that tripodal compounds that mimic this C-terminal region may function as DegP activators. Crystal structure of the Lpp trimer shows that all Tyr56 and Arg57 residues reside in the same plane (Fig. 1a)[23]. To mimic this planar structure, we synthesized tripodal peptidyl compounds, TMB_(Cys_peptide), in which three peptidyl arms are attached to benzylic positions of a scaffold, 1,3,5-trimethylbenzene (TMB), via the N-terminal cysteine (Fig. 1b and Supplementary Fig. 1 and Supplementary Table 1)[24,25].

**Compounds with three peptidyl arms are good DegP activators**. We initially tested compounds that contain various sizes of

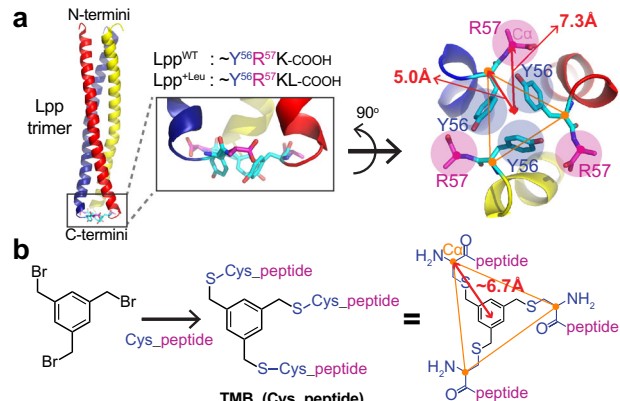

**Fig. 1 Design of the DegP-activating compounds mimicking Lpp^+Leu.** **a** Crystal structure of the Lpp trimer (PDB code, 1EQ7; cartoon presentation) with a close-up view of the C-terminal region. All Tyr56 (cyan sticks) and Arg57 (magenta sticks) residues roughly sit in the same plane. Distances from the center to the Tyr56-Cα and Arg57-Cα are 5.0 Å and 7.3 Å, respectively. **b** Synthesis of the tripodal peptidyl compounds, TMB_(Cys_peptide). Peptides are attached to the benzyl positions of 1,3,5-trimethylbenzene (TMB) via the N-terminal cysteine. Distance from the center to the cysteine-Cα in the extended tripodal compound is about 6.7 Å.

peptides from the Lpp^+Leu C-terminus, ~KYRKL, using an activation assay, in which we measured the cleavage rate of a poor model substrate, the reporter[14,22]. The reporter peptide is made of a short sequence from RseA and efficiently cleaved only in the presence of a good activator. Interestingly, activation was observed with TMB_CYRKL and TMB_CKYRKL, but not with TMB_CRKL or TMB_CKL (Figs. 2a, b). In particular, TMB_CYRKL increased the cleavage rate by at least ten times more than Lpp^+Leu and TMB_CKYRKL. Activation was followed by inhibition at higher concentrations of compounds and maximal activation occurred at around 100 μM. A previous report has shown that a good model substrate, 18–58, which is derived from the lysozyme, binds to both the active site and the PDZ1 domain of DegP, and allosterically activates DegP[14]. In an activation assay with the reporter peptide, 18–58 displayed two phases, an initial activation followed by inhibition, indicating competition for binding to the active site of DegP. The concentration-dependent activation and inhibition suggest that the tripodal compounds also bind to the active site of DegP. Alanine substitution of each residue of the pentapeptide (CYRKL) revealed that tyrosine is most important for activation (Fig. 2c).

Previously, we reported that an Lpp variant carrying YKI at the C-terminus (Lpp^YKI) instead of RKL in Lpp^+Leu has a better activation profile with its maximal activation at around 20 μM and a stronger growth inhibition at a heat-shock temperature[22]. We also tested the tripodal compound with the CYYKI arms (TMB_CYYKI). As expected, we obtained a similar biphasic curve with the maximal activation at around 10 μM, suggesting that this compound interacts with DegP more tightly than TMB_CYRKL (Fig. 2d, black circle). Alanine scanning of the peptide showed that the C-terminal isoleucine and two tyrosines are critical for the activation and tight binding of DegP (Fig. 2d). The importance of the residues was further confirmed by the comparison with TMB_CYYKL and TMB_CYRKI (Supplementary Fig. 2). Previous report showed that substrate binding accompanies several allosteric behaviors of DegP: proteolytic activation, assembly of cage-like proteolytic chamber and positively cooperative binding of a substrate[14]. Aside from the allosteric activation, we also analyzed the effects of the two compounds on assembly of cages and cooperative binding of a

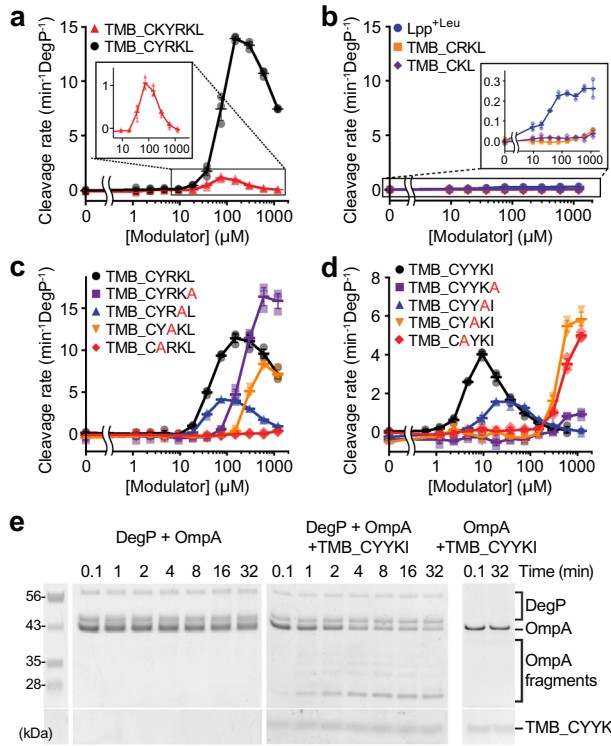

**Fig. 2 Tripodal peptidyl compounds activate the DegP protease.** Data are presented as dot plots with mean ±1 SD ($n = 3$ independent experiments). **a**, **b** Activation effects of compounds were monitored by measuring cleavage rates of the reporter peptide (100 μM) by DegP (1 μM for TMB_CKYRKL and TMB_CYRKL (**a**); 10 μM for Lpp$^{+Leu}$, TMB_CRKL, and TMB_CKL (**b**)) with increasing concentrations of Lpp$^{+Leu}$ or tripodal compounds. The insets show close-up views. **c** Effect of alanine substitution on each position of the pentapeptide (CYRKL) for DegP activation. **d** DegP activation effects of TMB_CYYKI or its variants with an alanine substitution were monitored as described in Fig. 2a. **e** Degradation of denatured OmpA (2 μM) by DegP (5 or 0 μM) was monitored by SDS-PAGE in the absence or presence of TMB_CYYKI (10 μM).

substrate, and found that both TMB_CYRKL and TMB_CYYKI assemble DegP cages, but that they differently affect the binding of model substrates (Supplementary Fig. 3).

Next, we tested whether the tripodal compound promoted degradation of a protein substrate, OmpA, which is an abundant outer membrane protein. Denatured OmpA was degraded not by DegP alone, but by TMB_CYYKI-activated DegP, indicating that the compound induces degradation of a protein that is not efficiently recognized and degraded by DegP (Fig. 2e and Supplementary Fig. 4). Good substrates can activate DegP, but their activation effects disappear when they are completely cleaved[14]. By contrast, TMB_CYYKI functions as a permanent activator; it remained almost intact and still activated DegP after longer incubation times (1–4 h) with DegP (Supplementary Fig. 5). Taken together, we demonstrated that the tripodal compounds with pentapeptide arms that mimic the Lpp variants are permanent activators of DegP.

**Tripodal compounds promote DegP-dependent growth inhibition.** Next, we tested the cellular effects of these tripodal compounds, TMB_CYRKL and TMB_CYYKI. Outer membrane (OM) of gram-negative bacteria often functions as a permeability barrier to molecules that are bigger than 600 Da[26]. As expected, TMB_CYRKL and TMB_CYYKI, whose molecular weights are ~2 kDa, did not inhibit growth of wild-type *Escherichia coli*

(W3110; Supplementary Fig. 6a). However, they inhibited growth of an OM-permeable mutant strain, the *imp4213* strain that is frequently used to study OM-impermeable antibiotics[27], with minimum inhibitory concentrations (MICs) of 320–640 μM and 80 μM, respectively (Fig. 3a and Supplementary Fig. 6b). Surprisingly, growth inhibition was observed at 30 °C, at which DegP is not essential for bacterial survival and Lpp$^{YKI}$ does not trigger growth defects[22]. These results suggest that the tripodal compounds turn a temperature-dependent essential protease into a temperature-independent toxic enzyme and that they have stronger cellular activity than the Lpp variants.

To obtain time-kill curves, we incubated cells at log-phase with compounds at 2× MICs and counted viable cells at several time points. Notably, TMB_CYRKL and TMB_CYYKI reduced the number of viable cells by two orders of magnitude at both 30 °C and 42 °C (Fig. 3b and Supplementary Fig. 6c). The *degP*-deletion strain (*imp4213 ΔdegP*), which does not grow at a high temperature (42 °C; Supplementary Fig. 6d), grew normally at 30 °C when treated with the compounds, indicating that their cellular toxicity requires the DegP function (Fig. 3c). To further confirm that the toxicity is dependent of DegP, we inserted into the *imp4213 ΔdegP* strain the plasmid that expresses either wild-type DegP (DegP$^{WT}$) or a catalytically inactive variant (DegP$^{S210A}$) in the presence of arabinose. The ectopic expression of DegP$^{WT}$, not DegP$^{S210A}$, suppressed the heat shock stress at a high temperature (42 °C; Figs. 3d, e). When we treated these two strains with TMB_CYYKI at 30 °C in the absence or presence of arabinose, we observed that TMB_CYYKI inhibited growth of the strain expressing DegP$^{WT}$, but not DegP$^{S210A}$ (Fig. 3f), suggesting that the toxicity of TMB_CYYKI requires catalytically active DegP.

It is unknown what proteins the over-activated DegP mainly degrades. Because DegP does not normally degrade folded proteins and most proteins in the periplasm and OM are exported as unfolded proteins across the cytoplasmic membrane after ribosomal synthesis[28], the newly synthesized and translocated proteins may become more accessible targets for the DegP over-proteolysis than the folded periplasmic proteins. If the over-activated DegP becomes toxic by degradation of folded proteins, the activator compounds would reduce cell viability regardless of whether cells are actively growing or not. We tested this idea by combining TMB_CYYKI with a ribosome inhibitor, chloramphenicol. Chloramphenicol alone (20 μg/ml) maintained the number of viable cells roughly at the same level for 12 h, and interestingly, TMB_CYYKI in a combination of chloramphenicol also showed the similar static pattern without considerable reduction of cell numbers at both 30 °C and 42 °C (Fig. 3g, and Supplementary Fig. 7). Consistent with this result, no reduction of viable cells was observed in combination with other antibiotics, tetracycline and rifampicin that inhibit protein synthesis and RNA synthesis, respectively, as well as in a single treatment of TMB_CYYKI to cells at stationary phase at which protein synthesis is substantially reduced (Fig. 3h). These results suggest that the toxicity of TMB_CYYKI is mainly the result of excessive degradation of newly synthesized proteins.

Previously we reported that cells overexpressing a hyperactive DegP variant, DegP$^{R207P/Y444A}$, died uniformly with an elongated morphology at both higher and lower temperatures, whereas the *ΔdegP* cells were mainly shorter and smaller at lethal high temperatures (Fig. 4a)[16]. Cells treated with TMB_CYYKI showed both elongated and normal morphology at both 30 °C and 42 °C (Fig. 4b), and, as expected, TMB_CYYKI did not induce any morphological difference in the *ΔdegP* cells (Fig. 4c). We also quantified cell length and confirmed that, at 42 °C, the *ΔdegP* cells and the DegP$^{R207P/Y444A}$-overexpressing cells are smaller and much longer, respectively, than the wild-type cells (Fig. 4d). The

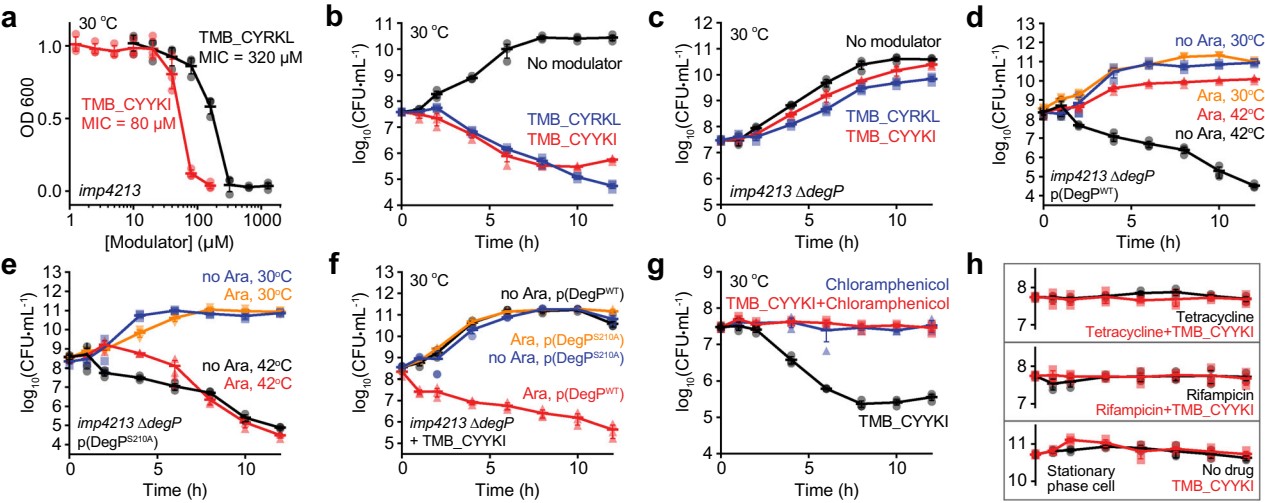

**Fig. 3 Tripodal compounds inhibit bacterial growth in a DegP-dependent fashion.** Data are presented as dot plots with mean ±1 SD ($n = 3$ independent experiments). **a** Growth of the *imp4213* cells in the presence of TMB_CYYKI or TMB_CYRKL. Overnight cultures in LB broth were diluted 100-fold in the presence of different concentrations of TMB_CYYKI or TMB_CYRKL. Growth was monitored by $OD_{600}$ after 12 h at 30 °C. **b** Time-kill curves of the *imp4213* strain with TMB_CYYKI, TMB_CYRKL or no compound. Cells at log phase ($OD_{600} = 0.01$) were incubated at 30 °C in the absence or presence of TMB_CYYKI (160 μM) or TMB_CYRKL (640 μM). Cells at different times were taken out to determine colony forming units (CFUs). **c** Time-kill curves of the *imp4213* Δ*degP* strain with TMB_CYYKI, TMB_CYRKL, or no compound at 30 °C. **d, e** Time-kill curves of the *imp4213* Δ*degP* strain that has a plasmid expressing either DegP$^{WT}$ (**d**) or DegP$^{S210A}$ (**e**) in the absence or presence of arabinose (Ara) at 30 °C or 42 °C. **f** Time-kill curves of the *imp4213* Δ*degP* strain containing a DegP$^{WT}$- or DegP$^{S210A}$-plasmid in the presence of TMB_CYYKI (2x MIC) at 30 °C. **g, h** Time-kill curves of the *imp4213* strain at 30 °C in the presence of TMB_CYYKI (160 μM), chloramphenicol (20 μg/ml), rifampicin (0.016 μg/ml), tetracycline (2 μg/ml), or both an antibiotic and TMB_CYYKI. Those of the cells at stationary phase in the absence or presence of TMB_CYYKI are also shown.

*imp4213* cells treated with TMB_CYYKI were slightly longer than those without TMB_CYYKI, while they were not as much longer as the DegP$^{R207P/Y444A}$-overexpressing cells (Fig. 4e). Although DegP$^{R207P/Y444A}$ and TMB_CYYKI-activated DegP appear to differently affect cells, these results suggest that TMB_CYYKI-driven growth inhibition is not the result of misfolded protein stress.

**Activators bind to the substrate-binding sites of DegP**. Two DegP-binding motifs of 18–58, a cleavage-site motif and a PDZ1-binding motif, bind to the active site region and the PDZ1 pocket of DegP, respectively (Supplementary Fig. 8a)[14]. To determine if the tripodal compounds bind to these substrate-binding sites or new allosteric sites, we solved the crystal structures of a DegP active-site mutant, DegP$^{S210A}$, with TMB_CYYKI or TMB_CYRKL at 3.6 Å and 4.2 Å resolution (PDB 6JJK and 6JJO), respectively, by molecular replacement using the peptide-deleted DegP dodecamer model (3OTP[14]; Table 1). Overall DegP structures were almost identical to that of the dodecameric DegP with an active conformation (rmsd 0.697 Å and 0.986 Å, respectively, for 2309 Cα atoms in six DegP subunits; Fig. 5a and Supplementary Fig. 8b). Additional electron density maps were found at positions where two degrons of 18–58 are present in the dodecamer model (Fig. 5b and Supplementary Fig. 8c). Because CYRKL and CYYKI have hydrophobic C-terminal residues that can occupy both the S1 pocket of the active site and the hydrophobic pocket of PDZ1, we believe that these peptides are located in the two substrate-binding sites in the same fashion as two degrons of 18–58. Although the lack of electron density for the central benzene ring hampers determination of the binding mode, we suggest that only one arm of the tripodal compounds in the co-crystal structures binds to either substrate-binding sites of DegP, because the distances between two cysteine-Cα atoms in the CYYKI peptides of the crystal structure are too long to justify

any of the three plausible modes for the bi- or tri-partite interactions (Supplementary Fig. 8d, 8e). Nevertheless, the simultaneous binding of multiple arms in solution cannot be ruled out, because a slight change of peptide conformation may allow the connection of two pentapeptides to the benzylic scaffold without changing the position of the two C-terminal hydrophobic residues (Supplementary Fig. 8f).

**The three-arm compound shows the highest activity**. To test whether less than three arms are sufficient for their function, we synthesized compounds with two CYYKI arms (DMB_CYYKI; DMB, 1,3-dimethylbenzene) or one arm (MMB_CYYKI; MMB, methylbenzene) (Fig. 6a). Several results suggested that the three-arm compound has the highest activity and mediates interactions different from one- or two-arm compounds. First, DMB_CYYKI and MMB_CYYKI had MIC values that are at least 10-fold higher than that of TMB_CYYKI (Supplementary Fig. 9). Second, they required higher concentrations for maximal activation, about 40 μM for DMB_CYYKI and 300 μM for MMB_CYYKI, indicating that their interactions for activation are 4-fold and 30-fold weaker, respectively, than that of TMB_CYYKI (Fig. 6b). Finally, isothermal titration calorimetry (ITC) experiments demonstrated a different binding stoichiometry of TMB_CYYKI from DMB_CYYKI and MMB_CYYKI. TMB_CYYKI showed similar stoichiometry ($N = 0.57$; Fig. 6c) with 18–58 ($N = 0.47$; Fig. 6d), whereas DMB_CYYKI and MMB_CYYKI showed the N values near 1 (0.98 for DMB_CYYKI and 1.11 for MMB_CYYKI; Figs. 6e, f). We also observed large peaks at low molar ratio of DMB_CYYKI and MMB_CYYKI. To further analyze this region, we titrated DMB_CYYKI in 0–0.4 molar ratio in a separate ITC experiment and found another N-value (0.22; Fig. 6g). Although it is unclear what this unusual N-value conveys, we did not see similar large peaks with TMB_CYYKI or 18–58. Because 18–58 binds to DegP at a 1:1 ratio in crystal structure, we believe that

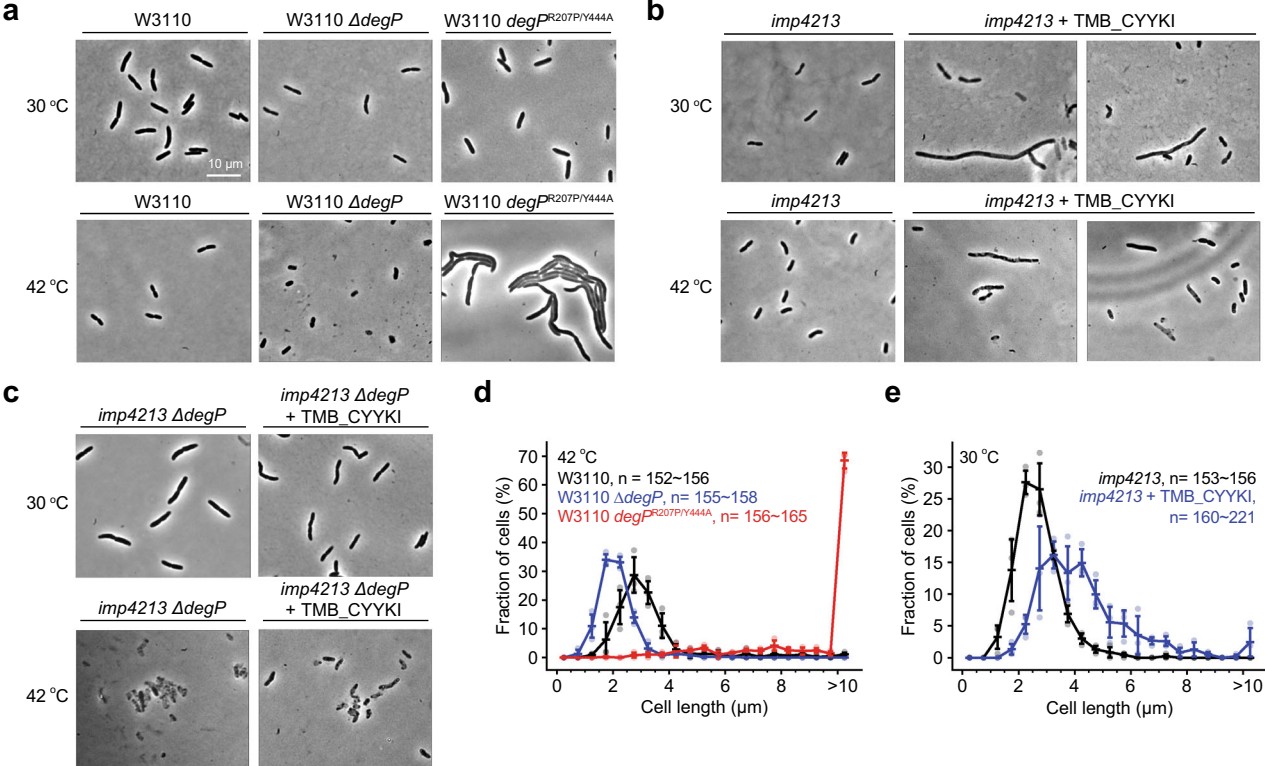

**Fig. 4 TMB_CYYKI induces an elongated morphology. a–c** Representative micrographs of *E. coli* W3110 (wild-type), Δ*degP*, or *degP*[R207P/Y444A] strains (**a**), the *imp4213* strain in the absence or presence of TMB_CYYKI (160 µM; **b**), or the *imp4213* Δ*degP* strain in the absence or presence of TMB_CYYKI (160 µM; **c**), at 30 °C or 42 °C after 4 h of incubation from $OD_{600} = 0.1$. **d, e** Quantification of cell length (*n* = 152–221) of the wild-type, Δ*degP*, or *degP*[R207P/Y444A] strains at 42 °C (**d**), or the *imp4213* strain at 30 °C in the absence or presence of TMB_CYYKI (160 µM; **e**). Data are presented as dot plots with mean ±1 SD (*n* = 3 independent experiments).

**Table 1 Data collection and refinement statistics (molecular replacement).**

|  | DegP[S210A] TMB_CYYKI (6JJK) | DegP[S210A] TMB_CYRKL (6JJO) |
|---|---|---|
| Data collection |  |  |
| Space group | C2 | C2 |
| Cell dimensions |  |  |
| *a, b, c* (Å) | 214.9, 122.4, 140.2 | 217.2, 123.5, 140.6 |
| *α, β, γ* (°) | 90, 117.7, 90 | 90, 118, 90 |
| Resolution (Å) | 50–3.60 (3.66–3.60)[a] | 50–4.20 (4.27–4.20)[a] |
| $R_{merge}$ (%) | 10.2 (52.7)[a] | 11.2 (66.5)[a] |
| *I*/σ*I* | 36.0 (5.9)[a] | 33.2 (5.9)[a] |
| Completeness (%) | 99.9 (100.0)[a] | 99.8 (100.0)[a] |
| Redundancy | 7.3 (7.5)[a] | 7.2 (7.1)[a] |
| Refinement |  |  |
| Resolution (Å) | 35–3.60 | 35–4.20 |
| No. reflections | 37,191 | 24,436 |
| $R_{work}$/$R_{free}$ | 19.9 / 26.5 | 21.6 / 27.8 |
| No. atoms | 16,997 | 16,907 |
| Protein | 16,997 | 16,907 |
| Ligand/ion | 0 | 0 |
| Water | 0 | 0 |
| *B*-factors | 126.0 | 151.78 |
| Protein | 126.0 | 151.78 |
| Ligand/ion | 0 | 0 |
| Water | 0 | 0 |
| R.m.s. deviations |  |  |
| Bond lengths (Å) | 0.003 | 0.003 |
| Bond angles (°) | 0.61 | 0.58 |

[a]Values in parentheses are for the highest-resolution shell.

the ITC result reflects not the enthalpy change of the 18–58 binding itself, but that of the overall conformational changes that are triggered by the 18–58 binding. Despite these differences, all the peptidyl compounds showed very similar effects on DegP cage

assembly: Half-maximal assembly required 1–2 µM of 18–58, TMB_CYYKI, or DMB_CYYKI, or only 7 µM MMB_CYYKI (Fig. 6h), indicating that a single peptidyl arm is sufficient for cage assembly. Taken together, the crystal structures suggest that the tripodal compounds mainly bind to the substrate-binding sites of DegP. Although only one peptidyl arm appears to interact with DegP in the crystal structure, three peptidyl arms are required for optimal activity with a binding mode different from those of one- or two-arm compounds.

In conclusion, our results demonstrate that over-activation of the DegP protease can be a new approach to find antibacterial drugs (Fig. 7). Unlike the previously reported activators[14,29], the tripodal compounds can inhibit bacterial growth and activate DegP without their own degradation. Although allosteric inhibitors of enzymes have been suggested to be of high priority for drug discovery against proteases[6], allosteric activators of proteases with antibiotic effects are shown only with our compounds and ADEPs. Our results also suggest that, while the conventional approach in antibiotic discovery is to inhibit essential proteins or pathways, targeting a non-essential enzyme by over-activation can also be an effective strategy. Although OM impermeability, high MICs, and the peptidyl nature of the tripodal compounds may limit the clinical development of our compounds, our established assays can serve as a tool to help screen chemical libraries to identify activator compounds that are more suitable for clinical development. The information we obtained concerning the binding mode of the compounds suggests that the activator molecules should target the substrate-binding sites of DegP. Moreover, because a mammalian homolog, HtrA2/Omi, functions in mitochondrial protein quality control and is associated with neurodegeneration and aging[30,31],

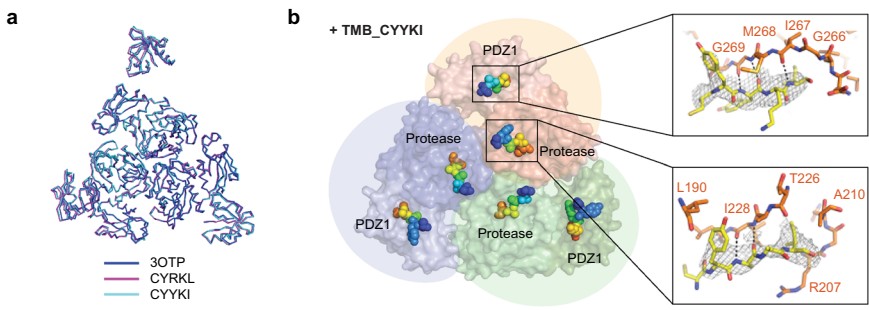

**Fig. 5 Co-crystal structures reveal that the tripodal compounds mainly bind to substrate-binding sites of DegP. a** Superposition of three trimers from the previously reported dodecameric structure (PDB code, 3OTP) and our structures with TMB_CYRKL or TMB_CYYKI. **b** Pentapeptides in DegP^S210A•TMB_CYYKI are shown in spheres (rainbow colors from the blue N-terminus to the red C-terminus) on the two substrate-binding sites of DegP. Three DegP monomers are colored differently in surface presentation. Two insets show close-ups of peptapeptides (yellow sticks with electron density maps) and nearby DegP residues (orange sticks) at the PDZ1 pocket (above) and the active-site region (below) of DegP.

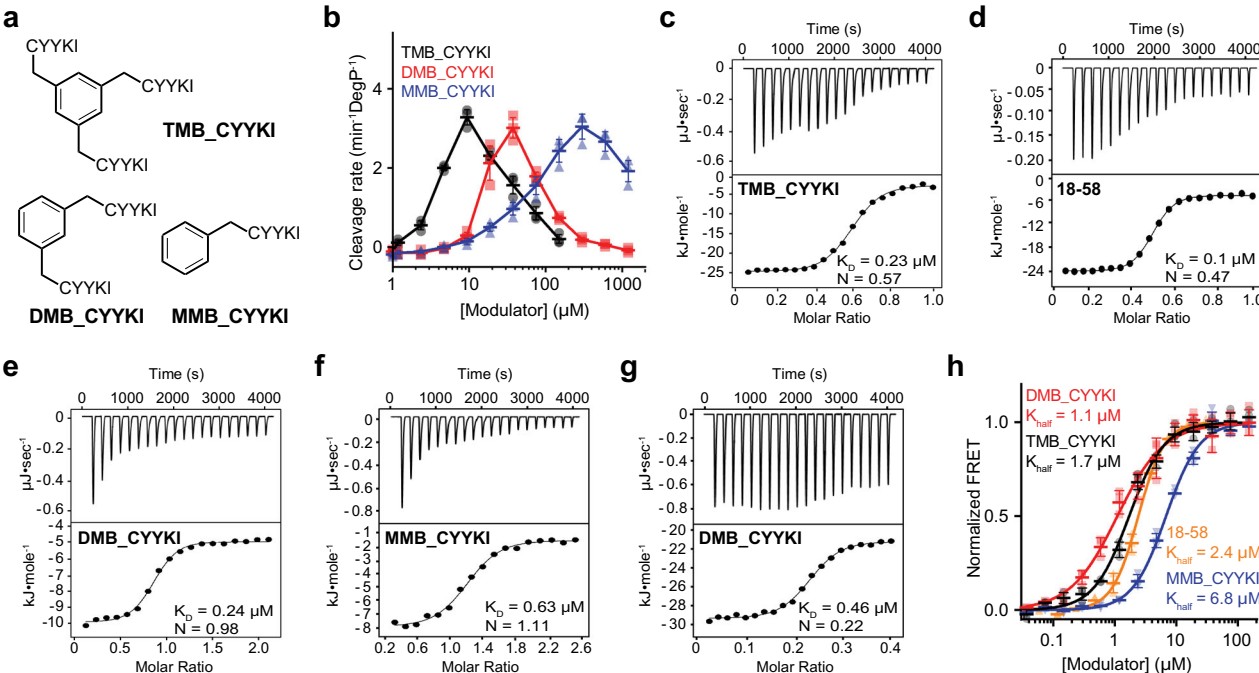

**Fig. 6 The compound with three arms has the best activity and properties different from those with one or two arms. a** Structures of compounds with three, two, or one peptidyl arm(s) (TMB_CYYKI, DMB_CYYKI, or MMB_CYYKI, respectively). **b** Activation effects of TMB_CYYKI, DMB_CYYKI, and MMB_CYYKI were monitored by measuring cleavage rates of the reporter peptide (100 µM) by DegP (1 µM) with increasing concentrations of compounds. Data are presented as dot plots with mean ± 1 SD (n = 3 independent experiments). **c–g** Isothermal titration calorimetry (ITC) experiments of TMB_CYYKI (**c**), 18–58 (**d**), DMB_CYYKI (**e, g** for 0–2 and 0–0.4 molar ratio, respectively), and MMB_CYYKI (**f**) with DegP^S210A. Data were fitted to the single binding site model. **h** DegP assembly was monitored by measuring fluorescence resonance energy transfer (FRET) of the donor- and acceptor-labeled DegP variants (0.1 M total) with increasing concentrations of 18–58, TMB_CYYKI, DMB_CYYKI, or MMB_CYYKI. Data were fitted to the Hill equation. Data are presented as dot plots with mean ± 1 SD (n = 3 independent experiments).

it is also of interest to find HtrA2 activators that may help treat associated human diseases either by restoring mitochondrial homeostasis or by disrupting mitochondrial integrity, as shown by activators of mitochondrial ClpP for killing cancer cells[32].

## Methods

**Bacterial strains and growth conditions.** Bacterial strains and plasmids are listed in Supplementary Tables 2 and 3, respectively. *E. coli imp4213* (RFM795; CGSC#: 14179), which is streptomycin-resistant, was obtained from *E. coli* genetic stock center at Yale University. The *imp4213 ΔdegP* strain was constructed by P1 transduction[33], in which the *ΔdegP::pheS-kan^R* allele from the *E. coli* SK322 donor strain was transferred to *E. coli imp4213* recipient strain. *E. coli* was grown in Luria-Bertani (LB) medium with aeration (200 rpm) at 37 °C overnight and diluted 1:100 in fresh LB. DegP^WT and its variants were prepared as previously described[14].

**Peptide synthesis.** Peptides were synthesized by solid phase peptide synthesis (SPPS) as previously described[34]. Wang resin and Fmoc protected amino acids (GL Biochem) were used in SPPS. First C-terminal amino acid (5 equiv.) and 4-Dimethylaminopyridine (DMAP, 0.1 equiv.; Sigma-Aldrich) dissolved in DMF were mixed with Wang resin for 20 min, and then N,N′-Diisopropylcarbodiimide (DIC, 4 equiv.; TCI) was added. The reaction mixture was incubated at room temperature for 40 h. Beads were washed three times with DMF and DCM. Acetic anhydride (10 equiv.; Sigma-Aldrich) and DMAP (0.1 equiv.) dissolved in DCM were added to beads for capping for 1 hour, and then the Fmoc group was removed using the deprotection solution (20% (v/v) piperidine (Alfa Aesar) in DMF) for 20 min after washing. The remaining amino acids were attached by coupling (5 equiv. HATU (GL Biochem) and 10 equiv. N,N-Diisopropylethylamine (Samchun) in DMF for 30 min), capping (Acetic anhydride: N,N-Diisopropylethylamine: DMF in a 5: 2.5: 92.5 ratio for 7 min), and deprotecting reactions. Coupling and deprotection were monitored by TNBS test. The global deprotection of all the protecting groups and the cleavage from the beads were

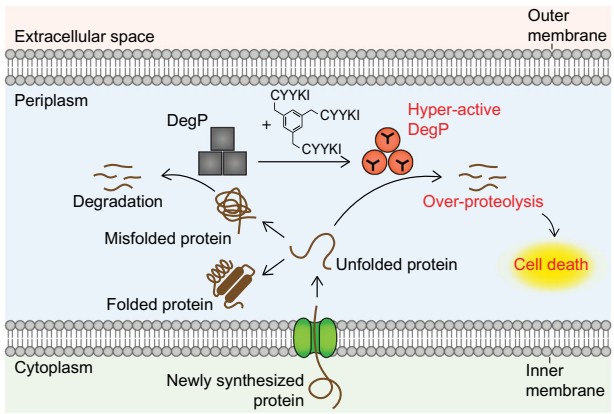

**Fig. 7 A model for bacterial killing by DegP over-activation.** DegP mainly degrades misfolded proteins, but its over-activation by the tripodal compounds triggers excessive degradation of newly synthesized proteins and leads to bacterial death.

performed with a cleavage cocktail (Trifluoroacetic acid (Samchun): water: 1,2-Ethanedithiol (Sigma-Aldrich): Triisopropylsilane (Sigma-Aldrich) in a 940:25:25:10 ratio for 2 h). The peptides were concentrated by solvent evaporation using nitrogen or air blowing and then precipitated by cold ether/hexane (1:1). Precipitates were dried and dissolved in water containing 0.1% TFA. Peptides were purified by reverse-phase high-performance liquid chromatography (RP-HPLC, Agilent 1260) and analyzed by matrix-assisted laser desorption/ionization (MALDI, Bruker) or LC-MS (Agilent 1260 infinity).

**Activator synthesis**. The 1,3,5-Tris(bromomethyl)benzene (10 µmol; Sigma-Aldrich) in acetonitrile (470 µL; Fisher) was added dropwise to a solution of the peptide (35 µmol) in NaHCO$_3$ (100 mM pH 8, 700 µL; Duksan). The reaction mixture was stirred for 1 h at room temperature. Product (TMB_(Cys_peptide)) was purified by HPLC and analyzed by MALDI. Conjugation of thiol, not amine, was confirmed by the absence of alkylation of the compounds with iodoacetamide. DMB_CYYKI and MMB_CYYKI were synthesized by using 1,3-bis(bromomethyl) benzene and benzyl bromide (Sigma-Aldrich), respectively.

**DegP activation assay with the reporter peptide**. DegP activation experiments were performed as previously described[14,22]. All the in vitro assays were performed in a buffer containing 50 mM sodium phosphate (pH 8) and 100 mM NaCl. Two-fold serial dilutions of the activator were incubated with DegP$^{WT}$ (1, 3, or 10 µM) for 5 min. After incubation, the reporter peptide (Abz-KASPVSLGY$^{NO2}$D; Abz, 2-aminobenzoic acid; Y$^{NO2}$, 3-nitrotyrosine; 100 µM final) was added. DegP activation was monitored by increased fluorescence by using a micro-plate reader (excitation 320 nm, emission 430 nm; Infinite F200pro, Tecan). Assays were performed in triplicates.

**Activator degradation assay**. An activator (10 µM) was mixed with DegP (10 µM). Aliquots were taken after 0.17, 1, 2, or 4 h. Each aliquot of the reaction solutions was quenched by the addition of TFA (1% final). The amount of activator for each sample was quantified using HPLC. Water (0.05% TFA) and methanol were used as the HPLC buffer.

**OmpA degradation assay**. OmpA (2 µM final; 100-fold dilution from the 8 M urea solution) was added to the reaction solutions of DegP (5 µM) and activator (0 or 10 µM). Each aliquot of the reaction solutions taken after 0.1, 1, 2, 4, 8, 16, or 32 min at 37 °C was quenched by the addition of SDS (2% final). OmpA protein degradation was monitored by the SDS-PAGE.

**Fluorescence resonance energy transfer**. Fluorescence resonance energy transfer (FRET) assay was performed as previously described[14,22]. Equimolar amounts of fluorescence donor- and acceptor-labeled DegP$^{S210A}$ (total 0.1, 1 or 3 µM) was mixed in a buffer containing 50 mM sodium phosphate (pH 8) and 100 mM NaCl. Two-fold serial dilutions of activators were added to the DegP mixtures. Alexa Fluor 555 C2-maleinide and 647 C2-maleinide (life technologies) were used as fluorescence dyes. Cage assembly of DegP was monitored by increased FRET ratio using the micro-plate reader (excitation 520 nm, emission 570 or 670 nm, Infinite M200pro, Tecan). Assays were performed in triplicates.

**Fluorescence anisotropy**. Fluorescence anisotropy assay was performed as previously described[14,22]. Two-fold serial dilutions of activators were mixed with a fluorescence-labeled peptide ($^{flC}$18–58 or $^{fl}$45–58, 50 nM; Oregon Green 488 maleimide, Invitrogen) and DegP$^{S210A}$ (10 µM for $^{flC}$18–58; 80 µM for $^{fl}$45–58). Anisotropy was monitored by using a micro-plate reader (excitation 485 nm, emission 535 nm, Infinite F200pro, Tecan). The assay was performed in triplicates.

**Minimum inhibitory concentration**. The susceptibility of *E. coli* to activators was analyzed by determining the MICs. Two-fold serial dilutions of activators in LB broth were added to bacteria (OD$_{600}$ = 0.1) in 96-well plates. Plates were incubated with shaking at 30 or 42 °C. After 12 h of incubation, the MIC was defined as the lowest concentration of the activator to inhibit the growth of bacteria by using a micro-plate reader (Absorbance 600 nm, Infinite M200pro, Tecan). The assay was performed in triplicates.

**Time-kill experiment**. Time-kill experiment was performed by counting the viable bacteria at each time point. The *E. coli imp4213* strain was cultured with streptomycin overnight and diluted 1:100 in fresh LB broth. When the OD$_{600}$ value became 0.01, the activator was added at 2x MIC. The sample was incubated with shaking (180 rpm) at 30 °C or 42 °C, and aliquots were taken after 0, 1, 2, 4, 6, 8, 10, or 12 h. These aliquots were serially diluted (ten-fold) and plated on LB agar plates for colony counts. Experiments were performed in triplicates. The kill curves were plotted with the CFU/mL against time.

Experiments with chloramphenicol (20 µg/ml, 10x MIC; Bio basic), tetracycline (2 µg/ml, 2x MIC; Bio basic), rifampicin (0.016 µg/ml, 2x MIC; Sigma-Aldrich) were performed in the same way. Experiments with cells in the stationary phase, however, were performed without dilution of the overnight culture.

**Imaging**. The W3110 (WT) and *imp4213* cells were grown overnight in LB broth at 37 °C and W3110 *ΔdegP*, *imp4213 ΔdegP* and W3110 *degP$^{R207P/Y444A}$* cells were grown at 30 °C. Cells were diluted to OD$_{600}$ = 0.1. The activator was added in the final concentration of 2x MIC. The sample was incubated with shaking (180 rpm) at 30 °C or 42 °C. After 4 h, 20 µL of the cell culture was placed between a coverslip and a 1.5% low-melting-temperature agarose gel pad (Agarose LE). Samples were placed on an inverted microscope (Olympus, IX-71) with a ×100 oil-immersed objective lens (Olympus). Phase contrast images were acquired using a cooled EMCCD camera (Andor iXon DU897). Metamorph software (Molecular Devices) was used to find the focus and control the measurements. For length quantification, a few hundred cells were taken from three independent replicate experiments (W3110, *n* = 153, 152, and 156; W3110 *ΔdegP*, *n* = 155, 155, and 158; W3110 *degP$^{R207P/Y444}$*, *n* = 158, 156, and 165; *imp4213*, *n* = 156, 153, and 156; *imp4213* + TMB_CYYKI, *n* = 173, 221, and 160).

**Crystallization and data collection**. DegP$^{S210A}$ (198 µM) and an activator (TMB_CYRKL or TMB_CYYKI; 297 µM) were incubated in a buffer containing 5 mM phosphate (pH 8.0) for 1 h at 295 K. Crystals of DegP$^{S210A}$ with TMB_CYRKL were grown using the hanging-drop vapor diffusion method by mixing equal volume (1 µL) of the protein-activator solutions and the reservoir solutions. A reservoir solution consisting of PEG3350 (11%) and tacsimate (0.1 M, pH 3.5) was used to grow crystals of DegP$^{S210A}$ with TMB_CYRKL. Crystals reached maximum size within 7 days at 295 K. Crystals of DegP$^{S210A}$ with TMB_CYYKI were grown, using a reservoir solution with PEG3350 (10%) and tacsimate (0.1 M, pH 3.4). The crystals were soaked in Paratone-N (Hampton Research) and incubated for 15 min before being flash-frozen in a nitrogen stream at 100 K. Native data for the DegP$^{S210A}$ with TMB_CYRKL or TMB_CYYKI were collected at the 5 C beamline and 7 A beamline of Pohang Accelerator Laboratory (Pohang, South Korea), respectively. The detector was an ADSC QUANTUM 270 and ADSC QUANTUM 315r (ADSC, USA), respectively, using a single wavelength of 0.9790 Å. The raw data were processed and scaled using the program suite HKL2000[35].

**Structure determination and refinement**. The structures of the DegP$^{S210A}$ with each activators were solved by the molecular replacement method using the hexamer model (chain A-F) of DegP$^{S210A}$ from *E. coli* (PDB ID: 3OTP)[14]. The original 3OTP structure contained a peptide substrate, and the peptide was removed for the calculation of molecular replacement. A cross-rotational search followed by a translational search was performed using the Phaser program[36]. Subsequent manual model building was carried out using the COOT program and restrained refinement was performed using the PHENIX program[37]. Several rounds of model building, simulated annealing, positional refinement with NCS restraints, and individual *B*-factor refinement were performed. Two structures determined in this study display Ramachandran statistics with 95.1% and 96.8% of residues in the most favored regions and 4.9% and 3.1% of residues in the allowed regions of the Ramachandran diagram, respectively. Table 1 lists the refinement statistics. The crystallographic asymmetric unit contained two trimers of DegP$^{S210A}$ with each modulator. Structural figures were drawn with PyMOL.

The atomic coordinates and structure factors for the DegP$^{S210A}$•TMB_CYYKI and DegP$^{S210A}$•TMB_CYRKL complexes were deposited in Protein Data Bank (PDB ID code 6JJK and 6JJO respectively).

**Isothermal titration calorimetry**. ITC experiments were performed using Affinity ITC instruments (TA Instruments, New Castle, DE, USA) at 298 K. DegP$^{S210A}$ mutant (18.10 μM for 0–2 molar ratio and 109.33 μM for 0–0.4 molar ratio), which were prepared in a buffer containing 5 mM phosphate (pH 8.0) were degassed at 295 K prior to measurements. Using a micro-syringe, 2.5 μL of DMB_CYYKI (300 μM) was added at intervals of 200 s to the DegP solution in the cell with gentle stirring. As for other ligands, 2.5 μL of MMB_CYYKI (453 μM), TMB_CYYKI (218.46 μM) or 18–58 substrate peptide (80 μM) was added at intervals of 200 s to the DegP solution (27.33 μM, 37.45 μM, and 20 μM, respectively). Data were fitted to the single binding site model. In experiments with 0–2 and 0–2.6 molar ratio of DMB_CYYKI and MMB_CYYKI, respectively, the data with 0–0.3 molar ratio were excluded for the fitting.

**Statistics and reproducibility**. Data in graphs with error bars are presented as means ± standard deviation (S.D.) from three independent experiments ($n = 3$). All Specific information on the number of replicates and statistical analyses are included in the figure legends. Data were processed with GraphPad Prism version 6.07.

**Reporting summary**. Further information on research design is available in the Nature Research Reporting Summary linked to this article.

## Data availability

The coordinates and structure factors of DegP$^{S210A}$•TMB_CYYKI and DegP$^{S210A}$•TMB_CYRKL have been deposited in the Protein Data Bank under the accession codes 6JJK and 6JJO, respectively. Source data for graphs in this study are available in Supplementary Data 1 (in excel format).

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

## Acknowledgements
The imp4213 strain (RFM795) was obtained from Coli Genetic Stock Center from Yale University. We thank Ga-eul Eom, Hyunbin Lee, Inseok Song, Daeje Seo, Heejin Roh for helpful discussions and technical assistance, and Yan Lee, Youngjun Song and Taeyang An for assistance in synthesizing compounds. We also thank the staffs of the 5C and 7A beamlines at the Pohang Light Source. This research was supported by Basic Science Research Program through the National Research Foundation of Korea (NRF) funded by the Ministry of Education (NRF-2014R1A1A2056722).

## Author contributions
S.K. and H.C. conceived and designed the work, analyzed and interpreted the data, and wrote the paper. H.C. performed the majority of experiments with helps from H.P. except for crystallographic analysis, ITC, and cell imaging. Y.C. and H.H.L. performed the crystallographic work. K.M. performed the ITC experiments. J.B.S. obtained the imaging data. All authors approved the paper.

## Competing interests

The authors declare no competing interests.
