## [Peer Review File · Communications Biology]

Reviewers' comments:

Reviewer #1 (Remarks to the Author):

In this work ("Over-activation of a Nonessential Bacterial Protease as an Antibiotic Strategy"), Cho and colleagues design small molecule that can cause allosteric activation of DegP and lead to growth inhibition due to this inappropriate activation. The authors further confirm the interaction between the compound and DegP by analyzing a co-crystal. The authors propose that antimicrobial over-activating non-essential pathways is an alternative to traditional antimicrobials inhibiting essential pathways. Overall, this work would be interesting to the protease, envelope biogenesis, and antibiotic discovery fields and is generally scientifically sound. However, there are several concerns that, if addressed, would increase the support for the model and strengthen the paper.

Major Concerns

1. Figure 2: In part 2a, data is included in the same graph with different reaction conditions. These data should be placed on a separate graph to make this clear. Also, in part 2B, it is not clear from the Figure legend and Materials and Methods whether the replicates were performed at the same time for the different compounds or not. If, in Figure 2B, all but the control compound were assayed simultaneously, this experiment needs to be repeated with the control compound.
2. The authors show that newly synthesized proteins are necessary for the effect of TMB_CYYKI and hypothesize that over-degradation of these proteins leads to the toxicity. However, as unfolded OMPs themselves can be toxic, it is important to show that there is actually a decrease in the amount of key OMPs or periplasmic proteins.
3. One important factor with a drug that does not target an essential function is the rate of resistance. What is the rate of degP- resistant mutants occurring at 30 C and 37 C in treated cultures?

Minor Comments

1. Ln 24-25: It would not be clear what is meant by "DegP-activating lipoprotein variants" to someone unfamiliar with DegP.
2. Ln 26: Is the permeable membrane necessary for compound entry or for synthetic activity with DegP activation?
3. Ln 36: "proteins, cell wall peptidoglycan,"
4. Ln 50-53: This section should mention that DegP can act as a chaperone at lower temperatures, although it acts as a protease at higher temperatures.
5. Ln 78/Figure 2: What is the reporter peptide (what protein does it come from)?
6. Figure 3B & S5C: It appears that with the CYYKI peptide treatment, the cell number starts to increase at 10 hours. Are there resistant mutants growing out? What mutation causes this, especially at 42 C where degP deletion is lethal?
7. Ln 152: I am not sure what "they" refers to ("although they may degrade...").
8. Ln 154: "degrade"
9. Figure 3e: The authors treat with chloramphenicol to demonstrate that protein synthesis is necessary to cause toxicity with TMB_CYYKI treatment. However, chloramphenicol (as a translation elongation inhibitor) also causes clogging and degradation of the Sec translocon and leads to envelope stress response activation which could alter the results (<https://www.ncbi.nlm.nih.gov/pmc/articles/PMC2832214/>). Therefore, the rifampicin data should be moved from Figure S8 to Figure 3, as it avoids this complication.
10. Ln 171: "Two degrons of 18-58" requires more explanation.
11. Binding of the compound to the substrate binding sites of DegP suggests the possibility of competitive inhibition. In a binding assay, does excess substrate compete the compound away from DegP?

Reviewer #2 (Remarks to the Author):

Cho et al report a novel antibiotic discovery strategy, by which cell growth of Gram-negative

bacteria (*E. coli*) can be inhibited by over-activating a non-essential periplasmic protease. Based on the structural basis for the Lpp-induced activation of protease DegP, authors synthesized a series of peptidomimetics and identified two compounds exhibiting appreciable activity for over-activating DegP *in vitro*. Then They demonstrated that these compounds could inhibit the cell growth of *E. coli* in a DegP-dependent manner. Furthermore, they determined the crystal structures of the DegP-peptidomimetic complexes, on the basis of which the peptidomimetics could be further optimized. Overall, this work is of interest. Nevertheless, I do have several concerns regarding the mechanism of this novel antibiotic discovery strategy and the conditions for bacterial cell killing assay.

Major concerns

- 1) Although the designed peptidomimetics were found to be able to over-activate DegP *in vitro*, authors have not provided any evidence to show that they do over-activate DegP in cells such that the cell growth is inhibited. Biochemical analysis of the protein levels of typical outer membrane proteins (e.g., OmpA, OmpF, OmpC, LamB) in cells, particularly the folded forms of these OMPs, may help to clarify this ambiguity.
- 2) Although the $\Delta degP$ mutant cells did not show growth arrest after treatment with the peptidomimetics, it is of interest to test whether the mutant carrying a S210A mutation in *degP*-encoding gene in *E. coli* genome is sensitive. Results may provide evidence to clarify question 1 and show that these compounds effect by activating the protease function of DegP *in vivo*.
- 3) These designed peptidomimetics have molecular weights of >2.3 kDa and thus only effect against the *E. coli* imp4213 mutant that has permeable outer membrane. This limitation may compromise the enthusiasm to this work. It is unknown whether small peptidomimetics (<600 Da) could be designed such that they are able to freely diffuse into the periplasmic for over-activating DegP.
- 4) Although peptidomimetics were further optimized that have higher affinity to DegP and also more effectively activate DegP *in vitro* (Fig. 5), it is disappointing that the did not teste whether these compounds could kill cells more effectively.
- 5) Authors stated the peptidomimetics were able to inhibit cell growth, but the data (Fig. 3) show that these compounds appear to have bactericidal activity rather than bacteriostatic activity. This point need to be clarified.

Minor issues.

- 6) Some spelling and grammar errors. The manuscript needs 'polishing' by a native speaker or professional editor.
- 7) What is the difference between agonist (activator) and antagonists (inhibitor) for DegP protease?

Reviewer #3 (Remarks to the Author):

"Over-activation of a Nonessential Bacterial Protease as an Antibiotic Strategy" by Cho et al is the first paper to depict the possibility of DegP, a non-essential bacterial periplasmic protease, to act as an antimicrobial when over-activated by synthetic C-termini analogs of DegP activators. The authors report allosteric activation of DegP by some tripodal peptidyl compounds that are C-termini analogs of different DegP activating lipoproteins. They show that the synthetic tripodal peptidyl modulators exert a DegP-mediated toxicity/growth inhibitory effect on an outer-membrane permeable *E. coli* mutant in a temperature independent fashion, whereas DegP is only essential at higher temperature for cell survival. The authors also describe that the peptidyl arms of the modulators bind to the substrate binding sites on DegP. They biochemically characterize the interaction between DegP and the activator molecules. The work is important because the findings suggest a novel antibiotic approach where DegP can be targeted for over-activation by synthetic

DegP modulators to inhibit bacterial growth. The work also provides information on the properties of the activator compounds that can be useful to design DegP activator drugs. The experiments seem to be appropriate for the questions and are well-performed. There are a couple places where controls or quantification of results are required. We also suggest some re-writes and rearrangements.

Major comments:

Line 101: Where are the data for controls without any added peptides? How do we know if this is activating, if you don't show the data for the untreated control? The control data must be shown at least in A, or provide a reasonable explanation for why you aren't showing it. Also, for Fig 2d, an OMP+TMB_CYYKI only control without DEgP would probably strengthen the conclusion. Cite any reference if this has been done before.

Lines 147-152 – it's not clear to me how the microscopy images can be used to make conclusions about misfolded proteins vs. over-proteolysis. This seems like a pretty indirect type of data for this conclusion. Are you trying to argue, that because the over-active DegP mutant and the treated cultures both have elongated cell, that the same thing must be going on in both of these? This is a stretch – there are many things that can cause cell elongation, and you can't conclude that the same proteins are being over-proteolyzed in both cases based purely on morphology. If you want to make any conclusions from the microscopy data, you will need to quantify cell lengths in the different conditions from a few hundred cells taken from 2-3 independent replicate cultures. Also, if your readers are meant to compare images in fig 3 and fig S6 to follow the logic of this conclusion, that is extra confusing. If these images are meant to be compared, please put them side-by-side in the same figure. If you don't want to quantify the microscopy, please change the language such that you don't draw conclusions from it.

Our suggestion would be to put all graphs and kill curves together in the main figure and move the microscopy to the supplement.

I don't understand why, in fig 5C and supplementary fig 11, the data for the DMB_CYYKI compound has been split into two parts. How was this done? Why is it not discussed in the results or methods sections? This seems highly suspect. There should be one graph for one experiment. If you did two experiments on DMB_CYYKI, please explain why, explain what you did differently, and discuss the results. If you did one experiment and you are separating the data after the fact – that is unacceptable. Put all the data on one graph and try your best to make sense of it, or don't show it at all.

Minor comments

- Please correct the numerous small grammatical errors. For example:

Line 35 – add "the" before majority

Line 37 – derivatives of old ...

Line 4 – mechanisms of action

And other errors with articles and verb tenses throughout the paper.

- Line 78 – can you please provide a bit more information in the results section about what this reporter is and how it works.

- Line 84-85 – please explain in what fashion 18-58 activates DegP – your readers need this background information to make sense of your data. It is not clear to us how the activation and inactivation pattern is explained by the binding motif.

- Line 171 – I don't understand what "degrons of 18-58" means. Please define obscure terms for

your readers.

- Lines 220-230. What is the cage assembly interaction? Please provide background information in the introduction that will allow your readers to understand this section. I suspect this background information will make other aspects of the paper more clear – is this also related to the “18-58” stuff? Which is mentioned repeatedly but never explained.
- Line 226-227 – this idea that assembly is decoupled from activation is confusing. Again, if you explain the general concepts in the introduction, that will help your readers make sense of this. But, I think you are making a conclusion based on your data here – and, since the relevant data is spread out across so many different figs and supp figs, it’s hard to follow the logic. Can you please explain clearly how you came to this conclusion, and call the relevant figures. Also, I think supplementary figs 11 and 12 should be in the body of the paper, as part of figure 5 – there is no reason to separate these out – we need to look at these all together to make sense of it.
- Minor re-writing at several places would clarify your conclusions more and support what your figures show. For example- Line 145 and 148: mention 30C after “grew normally with the compounds”, otherwise the fig does not match the sentence. Mention the temperature after “an elongated morphology”. Also, the figures say 43C but the text says 42C. May be it’s a typo?
- Line 27: Normal temperature means optimal temperature or relatively lower temperature? Be specific. A little more background on why you are doing experiments at 30C not 37C or so would have been helpful.
- Line 105: From fig 2c and sup fig 4, the pentapeptide compound is permanently activating DegP. So stronger or permanent instead of just “good” may be?
- Line 140: The temperature dependent essentiality of DegP is an important factor featured in this work, so rewriting the sentence may make it more clear e.g. “ temperature-dependent essential protease into a temperature independent toxic enzyme”.

Reviewers' comments:

Reviewer #1 (Remarks to the Author):

In this work (“Over-activation of a Nonessential Bacterial Protease as an Antibiotic Strategy”), Cho and colleagues design small molecule that can cause allosteric activation of DegP and lead to growth inhibition due to this inappropriate activation. The authors further confirm the interaction between the compound and DegP by analyzing a co-crystal. The authors propose that antimicrobial over-activating non-essential pathways is an alternative to traditional antimicrobials inhibiting essential pathways. Overall, this work would be interesting to the protease, envelope biogenesis, and antibiotic discovery fields and is generally scientifically sound. However, there are several concerns that, if addressed, would increase the support for the model and strengthen the paper.

Major Concerns

(#1-1) 1. Figure 2: In part 2a, data is included in the same graph with different reaction conditions. These data should be placed on a separate graph to make this clear. Also, in part 2B, it is not clear from the Figure legend and Materials and Methods whether the replicates were performed at the same time for the different compounds or not. If, in Figure 2B, all but the control compound were assayed simultaneously, this experiment needs to be repeated with the control compound.

=> As the reviewer suggested, Fig. 2a data are separated into two groups, depending on the enzyme concentration we used in the assay (see below for new Fig. 2).

As for Fig. 2b, the TMB_CYRKL data (gray) were actually collected together with those for other Ala-substituted compounds in Fig. 2b. Therefore, we left these data in Fig. 2b (color is changed to black), and added new data for TMB_CYRKL in Fig. 2a, which were obtained together with those for compounds in Fig. 2a. I want to recall that a part of the data for a single compound (different concentrations and replicates) might be collected simultaneously, but the data for a different compound were obtained in a separate experiment, which was usually done in the same day or the following day.

Figure 2 (new)

(#1-2) 2. The authors show that newly synthesized proteins are necessary for the effect of TMB_CYYKI and hypothesize that over-degradation of these proteins leads to the toxicity. However, as unfolded OMPs themselves can be toxic, it is important to show that there is actually a decrease in the amount of key OMPs or periplasmic proteins.

=> Unfolded OMPs can be toxic, but we showed several circumstantial evidences that the DegP over-proteolysis, not misfolded protein stress, causes the cell death in our experiments:

- (i) The TMB_CYYKI treatment in the $\Delta degP$ cells does not kill cells (Fig. 3c), but expression of WT DegP, not catalytically inactive DegP(S210A), from a plasmid in the $\Delta degP$ strain restores the toxicity (Fig. 3e; this new data were obtained related to the comment #2-2). These data are consistent with the idea of cell death by DegP over-proteolysis, but not with the idea of unfolded protein stress.
- (ii) The $\Delta degP$ strain dies due to the misfolded protein stress at heat shock condition (42°C or higher) and these cells are usually smaller than the wild-type cells, but cells treated with TMB_CYYKI are not smaller than the wild-type cells (Fig. 4b and 4d).
- (iii) TMB_CYYKI can kill cells without heat shock (30°C; Fig. 3b). TMB_CYYKI does not cause OMP misfolding enough to kill cells, because it does not kill the $\Delta degP$ cells (Fig. 3c).

As the reviewer suggested, the direct observation of decreasing level of proteins would help confirm our over-proteolysis hypothesis. We agree with this suggestion, but the practical problem is that there are quite a number of essential proteins that are required for cell survival in cell envelope: There are only two essential OMPs, BamA and LptD, which are the central components of the protein complexes that assemble OMPs and LPS, respectively, in OM. But, many other components in these two pathways (BamD, LptE, LptA, LptC, LptF, and LptG) are essential and exist in cell envelope. Many proteins in the biogenesis of OM lipoproteins (Lgt,

LspA, Lnt, LolA, LolB, LolC, and LolE) and in cell division (FtsQ, FtsL, FtsB, FtsW, and FtsI) are also essential and exposed to periplasm. Another critical problem is that DegP may degrade multiple non-essential proteins, which are synthetically lethal. Therefore, it is hard and very time-consuming to find what proteins the over-activated DegP degrades simply by monitoring the level of individual proteins.

(#1-3) 3. One important factor with a drug that does not target an essential function is the rate of resistance. What is the rate of degP- resistant mutants occurring at 30 C and 37 C in treated cultures?

=> As shown in the data below (for data at 42°C, see the response for the comment #1-9 below), we constantly observed that the TMB_CYYKI-treated cells eventually grow (left) and the re-grown cells do not show significant sensitivity to TMB_CYYKI any more (right). We sequenced the *degP* gene from three re-grown clones that are obtained from three independent experiments, but found no mutation at all. Therefore, we believe that mutations on other unknown genes are responsible for the toxicity suppression. A mutation that reduces the membrane permeability of the *imp4213* strain (e.g. a null-mutation in *bamB* (*yfgL*) was shown to be sufficient for the suppression of the membrane permeability of the *imp4213* strain (Ruiz et al. Cell 2005, 121, 307)) may be responsible for the suppression, but others are also possible, given the rather unfavorable properties of the compounds for drug development (OM impermeability, high MICs, and the peptidyl nature). Above all, our main point in this manuscript is not that we found a good lead compound for antibiotics, but that we found a novel target and strategy to develop new antibiotics. We think that the small molecule activators should be identified to carefully analyze the resistance occurrence in the DegP over-activation strategy.

Minor Comments

(#1-4) 1. Ln 24-25: It would not be clear what is meant by “DegP-activating lipoprotein variants” to someone unfamiliar with DegP.

=> We changed “DegP-activating lipoprotein variants” to “the lipoprotein variants that were previously reported to activate DegP” (line 21-22/27 in the manuscript without markup/with markup). Of note, the word limit (150 words) of abstract restricts a lengthy explanation on this part. A detailed description can be found in the first paragraph of the Results and Discussion section. To compensate the added words in this part, we shortened the other part, “over-activation of a periplasmic protease, DegP, which belongs to a highly conserved HtrA family of proteases”, to “over-activation of the periplasmic DegP protease, a member of the highly conserved HtrA family” in abstract (line 19-20/25-26). (modified parts are indicated with underlines.)

(#1-5) 2. Ln 26: Is the permeable membrane necessary for compound entry or for synthetic activity with DegP activation?

=> We believe that the permeable membrane is required at least for the compound entry, because the wild-type cells usually block the entry of molecules above 600 Da and did not show growth inhibition upon the addition of TMB_CYYKI or TMB_CYRKL. And it is still possible that the defect of the *imp4213* strain, the permeable OM, has the synthetic toxicity with the DegP activation triggered by our compounds. The *imp4213* strain contains a deletion in the *lptD* gene, which encodes an essential OMP that is important for LPS biogenesis, and the reduced LPS biogenesis is believed to make OM permeable. If the over-activated DegP degrades any protein in LPS biogenesis, the synthetic activity may inhibit bacterial growth.

Although we do not exclude a possibility of the synthetic activity, we still believe that the DegP over-activation alone can be toxic enough to kill bacteria. First, as we mentioned in the answer for the comment #1-2 above, all the proteins in LPS biogenesis are essential for cell viability, and thus over-proteolysis of any protein in this pathway could lead to cell death. Also, there are many other OM/periplasmic/IM proteins whose degradation would stop bacterial growth. Second, as we mentioned in the main text, we have previously reported that the overexpression of a hyperactive DegP variant, DegP(R207P/Y444A), could kill bacteria.

(#1-6) 3. Ln 36: “proteins, cell wall peptidoglycan,”

=> What we meant by that part is that the majority of antibiotics target (i) biosynthesis of nucleic acids, (ii) biosynthesis of proteins, (iii) biosynthesis of cell wall peptidoglycan, and (iv) membrane. To avoid confusion, we modified this part to “maintenance of membrane, and biosynthesis of nucleic acids, of proteins, and of cell wall peptidoglycan” (line 33-34/47-48).

(#1-7) 4. Ln 50-53: This section should mention that DegP can act as a chaperone at lower temperatures, although it acts as a protease at higher temperatures.

=> As the reviewer indicated, many reports about DegP mentioned that DegP can act as a chaperone at lower temperatures (e.g. 28°C) and as a protease at higher temperatures. However, I think that we need to carefully understand this argument with the following reasons:

- (i) First, the argument of dual functions as a chaperone and a protease came from the report by Spiess et al. (Cell 1999), mainly based on the refolding of MalS. However, there has been little/no other examples of the chaperone activity at lower temperature, although there has been numerous examples of the protease activity. For example, DegP is a very good protease at 23°C against good substrates (ref 14).
- (ii) Second, although there has been no clear molecular mechanism known to convert a chaperone to a protease in a temperature-dependent fashion, a clear molecular mechanism of DegP activation by *substrate-binding* has been reported after the Spiess paper (ref 12-14). When DegP binds to a good substrate, DegP activity could be enhanced by more than 100-fold even at 23°C (ref 14).
- (iii) Third, enzyme activities are usually higher at higher temperatures, as long as higher temperatures do not break protein folding. Thus, higher proteolytic activity of DegP at higher temperature is natural as an enzyme.

In my opinion, DegP activation is governed mainly by substrate-binding, which triggers the allosteric conformation change to an active form, and temperature is only a minor aspect of DegP activation. I also think that DegP may undergo a kind of a triage process when it recognizes a substrate: DegP is not fully active without proper substrate binding. But DegP constantly tries to find a good substrate via temporary binding and release of proteins, during which some proteins may escape and fold correctly (chaperone activity), but others are trapped and degraded by DegP (protease activity). The latter can also activate DegP. I think that this manuscript does not require any of this argument at all, and thus did not mention it.

(#1-8) 5. Ln 78/Figure 2: What is the reporter peptide (what protein does it come from)?

=> As the reviewer asked, we added one more sentence, “The reporter peptide is made of a short sequence from RseA and efficiently cleaved only in the presence of a good activator.” (line 77-78/126-127).

(#1-9) 6. Figure 3B & S5C: It appears that with the CYYKI peptide treatment, the cell number starts to increase at 10 hours. Are there resistant mutants growing out? What mutation causes this, especially at 42 C where degP deletion is lethal?

=> We performed the kill-curve experiments with TMB_CYYKI again 42°C (see a figure below; also see the response for the comment #1-3 for data at 30°C). Three cultures independently grown and treated with TMB_CYYKI (2x MIC) constantly showed a decrease of viable cells until ~10 hours and an increase between 10~24 hours (left). We took one colony from each culture, and they all grew fine with TMB_CYYKI at 42°C (right), indicating that DegP still reduces misfolded protein stress. We sequenced the *degP* gene in these three strains, and found no mutation. The suppression of the TMB_CYYKI toxicity may come from mutations in other unknown genes, as we discussed in the response for the comment #1-3 above.

(#1-10) 7. Ln 152: I am not sure what “they” refers to (“although they may degrade...”).

=> We modified the whole sentence to “Although DegP^{PR207P/Y444A} and TMB_CYYKI-activated DegP might degrade slightly different sets of proteins, these results are consistent with the idea that TMB_CYYKI inhibits bacterial growth not by misfolded protein stress but by over-proteolysis.” (line 164-166/279-281).

(#1-11) 8. Ln 154: “degrade”

=> We corrected it (line 138/221).

(#1-12) 9. Figure 3e: The authors treat with chloramphenicol to demonstrate that protein synthesis is necessary to cause toxicity with TMB_CYYKI treatment. However, chloramphenicol (as a translation elongation inhibitor) also causes clogging and degradation of the Sec translocon and leads to envelope stress response activation which could alter the results (<https://www.ncbi.nlm.nih.gov/pmc/articles/PMC2832214/>). Therefore, the rifampicin data should be moved from Figure S8 to Figure 3, as it avoids this complication.

=> As the reviewer suggested, we moved the data for tetracycline, rifampicin, and the stationary cells to Fig. 3f (see below for new Fig. 3).

Figure 3 (new)

(#1-13) 10. Ln 171: “Two degrons of 18-58” requires more explanation.

=> To avoid the word “degron”, we changed this part, “Two degrons of 18-58, a cleavage-site degron and a PDZ1-binding degron” to “Two DegP-binding motifs of 18-58, a cleavage-site motif and a PDZ1-binding motif” (line 176/291).

(#1-14) 11. Binding of the compound to the substrate binding sites of DegP suggests the possibility of competitive inhibition. In a binding assay, does excess substrate compete the compound away from DegP?

=> We believe that the two phases (activation and inhibition) in the activation assay (Fig. 2a-c) indicate the cooperation and competition, respectively, for substrate binding.

To directly test competition or cooperation of binding, we performed binding assay by measuring anisotropy of a fluorescent-labeled substrate peptide in a single concentration with increasing amounts of the tripod peptide compounds (Supplementary Fig. 3). We used two substrate peptides, one with two DegP-binding motifs (¹¹18-58) and another with one DegP-binding motif (¹¹45-58). We found that TMB_CYRKL shows both cooperation and competition depending on the activator concentration: lower concentrations of the compound (5~40 μM for ¹¹18-58 and 10~80 μM for ¹¹45-58) enhance the binding, and higher concentrations (> 40 μM for ¹¹18-58 and > 80 μM for ¹¹45-58) interrupt the interaction. However, TMB_CYYKI showed only competition: the substrate peptide binding decreases with increasing amounts of the compound. We believe that, at lower concentrations, the compound may occupy only a

fraction of the three active sites and allosterically change the DegP conformation to an active form, which has higher affinity to a model substrate. Therefore, substrates can bind to the empty active sites in the presence of the compound more tightly than without the compound (positive cooperativity of substrate binding). At higher concentrations, the compound occupies all three active sites, and thus competitively inhibits the substrate binding. We do not know why TMB_CYYKI only shows the competition mode in the binding assay, although it shows two phases in the activation assay. It may exclude the two model substrates we used in the binding assay, but not other protein substrates, because we observed the direct DegP activation by TMB_CYYKI in the OmpA degradation assay (Fig. 2d).

Reviewer #2 (Remarks to the Author):

Cho et al report a novel antibiotic discovery strategy, by which cell growth of Gram-negative bacteria (*E. coli*) can be inhibited by over-activating a non-essential periplasmic protease. Based on the structural basis for the Lpp-induced activation of protease DegP, authors synthesized a series of peptidomimetics and identified two compounds exhibiting appreciable activity for over-activating DegP *in vitro*. Then They demonstrated that these compounds could inhibit the cell growth of *E. coli* in a DegP-dependent manner. Furthermore, they determined the crystal structures of the DegP-peptidomimetic complexes, on the basis of which the peptidomimetics could be further optimized. Overall, this work is of interest. Nevertheless, I do have several concerns regarding the mechanism of this novel antibiotic discovery strategy and the conditions for bacterial cell killing assay.

Major concerns

(#2-1) 1) Although the designed peptidomimetics were found to be able to over-activate DegP *in vitro*, authors have not provided any evidence to show that they do over-activate DegP in cells such that the cell growth is inhibited. Biochemical analysis of the protein levels of typical outer membrane proteins (e.g., OmpA, OmpF, OmpC, LamB) in cells, particularly the folded forms of these OMPs, may help to clarify this ambiguity.

=> We provided several circumstantial evidences that show that the DegP over-activation by our compounds in cells causes cell growth inhibition:

- (i) As the reviewer mentioned, we clearly demonstrated that our compounds over-activate DegP *in vitro*, and if these compounds enter the cells, the most probable event is to activate DegP, too.
- (ii) Growth inhibition depends both on our compounds and the proteolytically active DegP. Our compounds could inhibit growth of the *imp4213* cells (OM-permeable cells), but not of the *imp4213 ΔdegP* cells (Fig. 3a-c). Also, the ectopic expression of wild-type DegP from a plasmid, but not of a catalytically inactive DegP (DegP_S210A), restored the growth inhibition in the *imp4213 ΔdegP* cells (Fig. 3d-e; also see the response for the comment **#2-2** below). Therefore, our compounds “do” something on DegP for bacterial growth inhibition, and of the two ways to inhibit cell growth—DegP over-proteolysis and DegP inactivation—the former is consistent with these data. This conclusion is also consistent with the data that the compound-treated cells are not smaller than the wild-type cells, although the *ΔdegP* cells with the misfolded protein stress are smaller.
- (iii) It has been already shown that the over-activated DegP could inhibit cell growth. The overexpression of the hyperactive DegP variant, DegP(R207P/Y444A), could kill bacteria (ref 16). Therefore, it is not surprising that DegP overactivation by our compounds also inhibit cell growth.

We agree that the direct observation of protein degradation in cells strengthens our argument. However, as we explained in the response for the comment #1-2, there are a number of essential proteins in cell envelope, which could be a target of DegP proteolysis. Furthermore, degradation of multiple non-essential proteins may also kill cells. Therefore, it is hard to find what proteins the activator-bound DegP degrades simply by checking the level of individual proteins. Although we do not know the target proteins of the activator-bound DegP (we mentioned in the main text), we provided enough evidences to support the idea that our compounds inhibit cell growth by DegP over-activation. We do not think that the information about the levels of the suggested OMPs (OmpA, OmpC, OmpC, LamB) helps to identify targets, because none of them are essential for cell viability, and thus their degradation would not lead to cell growth inhibition.

(#2-2) 2) Although the $\Delta degP$ mutant cells did not show growth arrest after treatment with the peptidomimetics, it is of interest to test whether the mutant carrying a S210A mutation in *degP*-encoding gene in *E. coli* genome is sensitive. Results may provide evidence to clarify question 1 and show that these compounds effect by activating the protease function of DegP in vivo.

=> As the reviewer suggested, we added the new data with the DegP(S210A) mutant (Fig. 3d-e; see the comment #1-12 for new Fig. 3). Although we were unable to obtain the *imp4213 degP(S210A)* strain, we used a plasmid to express either wild-type DegP or DegP(S210A). Wild-type DegP, but not DegP(S210A), expressed from a plasmid complemented the resistance against misfolded protein stress at heat shock condition (Fig. 3d). More importantly, wild-type DegP, but not DegP(S210A), restored the compound-dependent growth inhibition in the *imp4213 $\Delta degP$* cells (Fig. 3e).

We also add texts for these new figures (line 129-136/213-219), which is as follows: “To further confirm that the toxicity is dependent of DegP, we inserted into the *imp4213 $\Delta degP$* strain the plasmid that expresses either wild-type DegP (DegP^{WT}) or a catalytically inactive variant (DegP^{S210A}) in the presence of arabinose. The ectopic expression of DegP^{WT}, not DegP^{S210A}, suppressed the heat shock stress at a high temperature (42°C; Fig. 3d). When we treated these two strains with TMB_CYYKI at 30°C in the absence or presence of arabinose, we observed that TMB_CYYKI inhibited growth of the strain expressing DegP^{WT}, but not DegP^{S210A} (Fig. 3e), suggesting that the toxicity of TMB_CYYKI requires catalytically active DegP.”

(#2-3) 3) These designed peptidomimetics have molecular weights of >2.3 kDa and thus only effect against the *E. coli imp4213* mutant that has permeable outer membrane. This limitation may compromise the enthusiasm to this work. It is unknown whether small peptidomimetics (<600 Da) could be designed such that they are able to freely diffuse into the periplasmic for over-activating DegP.

=> We agree that these compounds have limited potential for antibiotic development, because they are not OM-permeable and may be metabolically unstable. However, our main point is not that we found a good lead compound for antibiotics, but that we found a novel target and strategy to develop new antibiotics: over-activation of a non-essential protease. This strategy is quite unusual because the current paradigm of antibiotic development is to target essential proteins or pathways by inhibition. This is a good proof-of-concept study for a new strategy, and thus can be of importance and of general interest, because the limited targets and strategies for antibiotic development have worsened the current crisis of antibiotic resistance. Many alternative strategies have been suggested, including targeting virulence, allosteric

inhibitors, bacteriophages, and microbiome modulation, and more new strategies should be explored to avoid the “post-antibiotic era”.

Definitely the next step is to identify small molecule activators of DegP. Initial test of a small set of small molecule library (~10,000) in our laboratory did not result in any hit of high potential, but given the activation assay we established, it would be interesting to see if we get any hits from larger libraries.

(#2-4) 4) Although peptidomimetics were further optimized that have higher affinity to DegP and also more effectively activate DegP in vitro (Fig. 5), it is disappointing that they did not test whether these compounds could kill cells more effectively.

=> We have tested the additional compounds (DMB_CYYKI and MMB_CYYKI). DMB_CYYKI and MMB_CYYKI were less active (higher MICs) than TMB_CYYKI (Supplementary Fig. 9). In fact, MICs cannot be determined at 30°C, because cell growth inhibition was not significant at the highest concentrations. Therefore, we could not perform time-kill experiments, in which we usually add compounds at 2x MIC. They also required higher concentrations for maximal activation than TMB_CYYKI (Fig. 6b). Therefore, they are less effective in DegP activation.

(#2-5) 5) Authors stated the peptidomimetics were able to inhibit cell growth, but the data (Fig. 3) show that these compounds appear to have bactericidal activity rather than bacteriostatic activity. This point needs to be clarified.

=> Minimum inhibitory concentration (MIC) is defined as the concentration of a compound that inhibits bacterial growth. Minimum bactericidal concentration (MBC) is defined as the concentration that results in a 1000-fold reduction of bacterial density (99.9% killed). The definition of a bactericidal antibiotic is that the compound has the MBC/MIC value ≤ 4 . Time-kill curves are not sufficient to determine whether the compound is bactericidal or bacteriostatic. We could not determine MBC because our compounds were not soluble in higher concentration than 2x MIC. Therefore, we did not mention whether our compounds are “bactericidal” or “bacteriostatic”.

Minor issues.

(#2-6) 6) Some spelling and grammar errors. The manuscript needs 'polishing' by a native speaker or professional editor.

=> As the reviewer suggested, our manuscript was further polished by a fluent English speaker who has an editing experience.

(#2-7) 7) What is the difference between agonist (activator) and antagonists (inhibitor) for DegP protease?

=> The data in this manuscript and our previous data suggest that the activation and inhibition is concentration-dependent: a compound can be an activator at lower concentrations but become an inhibitor at higher concentrations. The reason is that DegP is allosterically activated by substrate binding, and our activators bind to the substrate binding sites. The trimeric allosteric system allows that a partial occupancy of a substrate or an activator at three active sites (substrate binding sites) triggers a conformational change to an active form, so that the empty active sites can accept and efficiently cleave other substrates. However, these

sites are saturated at higher concentrations of a substrate or an activator, and few empty sites are available for binding and cleaving other substrates.

Reviewer #3 (Remarks to the Author):

“Over-activation of a Nonessential Bacterial Protease as an Antibiotic Strategy” by Cho et al is the first paper to depict the possibility of DegP, a non-essential bacterial periplasmic protease, to act as an antimicrobial when over-activated by synthetic C-termini analogs of DegP activators. The authors report allosteric activation of DegP by some tripodal peptidyl compounds that are C-termini analogs of different DegP activating lipoproteins. They show that the synthetic tripodal peptidyl modulators exert a DegP-mediated toxicity/growth inhibitory effect on an outer-membrane permeable E. coli mutant in a temperature independent fashion, whereas DegP is only essential at higher temperature for cell survival. The authors also describe that the peptidyl arms of the modulators bind to the substrate binding sites on DegP. They biochemically characterize the interaction between DegP and the activator molecules. The work is important because the findings suggest a novel antibiotic approach where DegP can be targeted for over-activation by synthetic DegP modulators to inhibit bacterial growth. The work also provides information on the properties of the activator compounds that can be useful to design DegP activator drugs. The experiments seem to be appropriate for the questions and are well-performed. There are a couple places where controls or quantification of results are required. We also suggest some re-writes and rearrangements.

Major comments:

(#3-1) Line 101: Where are the data for controls without any added peptides? How do we know if this is activating, if you don't show the data for the untreated control? The control data must be shown at least in A, or provide a reasonable explanation for why you aren't showing it. Also, for Fig 2d, an OMP+TMB_CYYKI only control without DEgP would probably strengthen the conclusion. Cite any reference if this has been done before.

=> As the reviewer suggested, we included the data without peptides (DegP + reporter), which are dots with $x = 0$ in Fig. 2a-c (see the comment #1-1 for new Fig. 2). As for Fig. 2d, we added a control, OMP+TMB_CYYKI, in which no OmpA degradation was observed. We also added all the images of full gels in Supplementary Fig. 4.

(#3-2) Lines 147-152 – it's not clear to me how the microscopy images can be used to make conclusions about misfolded proteins vs. over-proteolysis. This seems like a pretty indirect type of data for this conclusion. Are you trying to argue, that because the over-active DegP mutant and the treated cultures both have elongated cell, that the same thing must be going on in both of these? This is a stretch – there are many things that can cause cell elongation, and you can't conclude that the same proteins are being over-proteolyzed in both cases based purely on morphology. If you want to make any conclusions from the microscopy data, you will need to quantify cell lengths in the different conditions from a few hundred cells taken from 2-3 independent replicate cultures. Also, if your readers are meant to compare images in fig 3 and fig S6 to follow the logic of this conclusion, that is extra confusing. If these images are meant to be compared, please put them side-by-side in the same figure. If you don't want to quantify the microscopy, please change the language such that you don't draw conclusions from it.

Our suggestion would be to put all graphs and kill curves together in the main figure and move the microscopy to the supplement.

=> As the reviewer suggested, we performed the imaging experiments again, and quantified cell length of different strains or in different conditions (Fig. 4d). We added the corresponding text (line 160-163/275-278), as follows “We also quantified the cell length and confirmed that, at 42°C, the $\Delta degP$ cells and the $DegP^{R207P/Y444A}$ -overexpressing cells are smaller and much longer, respectively, than the wild-type cells (Fig. 4d, left). The $imp4213$ cells treated with TMB_CYYKI were slightly longer than those without TMB_CYYKI, while they were not as much longer as the $DegP^{R207P/Y444A}$ -overexpressing cells (Fig. 4d, right).”

We also toned down the conclusion from this imaging experiment, and mentioned that “Although $DegP^{R207P/Y444A}$ and TMB_CYYKI-activated $DegP$ may degrade a slightly different set of proteins, these results are consistent with the idea that TMB_CYYKI inhibits bacterial growth not by misfolded protein stress but by over-proteolysis.” (line 164-166/279-281). We put graphs and kill curves obtained at 30°C in Fig. 3, while those obtained at 42°C and a minor graph (MIC determination of antibiotics) are in Supplementary Fig. 6-7. We put all the imaging data in Fig. 4.

Figure 4 (new)

(#3-3) I don't understand why, in fig 5C and Supplementary fig 11, the data for the DMB_CYYKI compound has been split into two parts. How was this done? Why is it not discussed in the results or methods sections? This seems highly suspect. There should be one graph for one experiment. If you did two experiments on DMB_CYYKI, please explain why, explain what you did differently, and discuss the results. If you did one experiment and you are separating the data after the fact – that is unacceptable. Put all the data on one graph and try your best to make sense of it, or don't show it at all.

=> The data for DMB_CYYKI has not been split into two parts. We performed two ITC experiments with different concentrations of $DegP(S210A)$, which cover molar ratios of 0.3~2

(Fig. 6c, bottom left) and 0~0.4 (Fig. 6c, bottom center). We also added explanation why we did two experiments with DMB_CYYKI, which reads as follows: “... whereas DMB_CYYKI and MMB_CYYKI showed the N values near 1 (0.98 for DMB_CYYKI and 1.11 for MMB_CYYKI; Fig. 6c, bottom left and right). We also observed large peaks at low molar ratio of DMB_CYYKI and MMB_CYYKI. To further analyze this region, we titrated DMB_CYYKI in 0-0.4 molar ratio in a separate ITC experiment and found another N-value (0.22; Fig. 6c, bottom center). Although it is unclear what this unusual N-value conveys, we did not see similar large peaks with TMB_CYYKI or 18-58.” (line 219-224/347-360).

We also moved all ITC data to Fig. 6c (see below for new Fig. 6).

Figure 6 (new)

Minor comments

(#3-4) • Please correct the numerous small grammatical errors. For example:

Line 35 – add “the” before majority

Line 37 – derivatives of old ...

Line 4 – mechanisms of action

And other errors with articles and verb tenses throughout the paper.

=> We corrected what the reviewer suggested: from “the majority of” to “most” (line 32/46), from “derivatives with” to “derivatives of” (line 34/48), and from “mechanism of actions” to “mechanisms of action” (line 37/51). We also polished the text with the help of a fluent English speaker.

(#3-5) • Line 78 – can you please provide a bit more information in the results section about what this reporter is and how it works.

=> (From the response for the comment #1-8) As the reviewer asked, we added one more sentence, “The reporter peptide is made of a short sequence from RseA and efficiently cleaved only in the presence of a good activator.” (line 77-78/126-127).

(#3-6) • Line 84-85 – please explain in what fashion 18-58 activates DegP – your readers need this background information to make sense of your data. It is not clear to us how the activation and inactivation pattern is explained by the binding motif.

=> As the reviewer suggested, we further explained the 18-58 peptide. This part now reads as follows: “A previous report has shown that a good model substrate, 18-58, which is derived from the lysozyme, binds to both the active site and the PDZ1 domain of DegP, and allosterically activates DegP. In an activation assay with the reporter peptide, 18-58 displayed two phases, an initial activation followed by inhibition, indicating competition for binding to the active site of DegP. The concentration-dependent activation and inhibition suggests that the tripodal compounds also bind to the active site of DegP.” (line 82-87/131-136).

(#3-7) • Line 171 – I don’t understand what “degrons of 18-58” means. Please define obscure terms for your readers.

=> (From the response for the comment **#1-13**) To avoid the word “degron”, we changed this part, “Two degrons of 18-58, a cleavage-site degron and a PDZ1-binding degron” to “Two DegP-binding motifs of 18-58, a cleavage-site motif and a PDZ1-binding motif” (line 176/291).

(#3-8) • Lines 220-230. What is the cage assembly interaction? Please provide background information in the introduction that will allow your readers to understand this section. I suspect this background information will make other aspects of the paper more clear – is this also related to the “18-58” stuff? Which is mentioned repeatedly but never explained.

=> The substrate binding triggers proteolytic activation and assembly of bigger oligomers (usually characterized as cage-like 12-mer and 24-mer). However, we found that “cage-assembling interaction” is confusing and unnecessary in this sentence. Therefore, we modified to “cage assembly” (line 229/366).

(#3-9) • Line 226-227 – this idea that assembly is decoupled from activation is confusing. Again, if you explain the general concepts in the introduction, that will help your readers make sense of this. But, I think you are making a conclusion based on your data here – and, since the relevant data is spread out across so many different figs and supp figs, it’s hard to follow the logic. Can you please explain clearly how you came to this conclusion, and call the relevant figures. Also, I think Supplementary figs 11 and 12 should be in the body of the paper, as part of figure 5 – there is no reason to separate these out – we need to look at these all together to make sense of it.

=> Here, we also found that the comment, “assembly is decoupled from activation”, is not necessary at all. Therefore, we deleted this comment (line 229/366). As the reviewer suggested, we also moved Supplementary Fig. 11 and 12 to Fig. 6c and 6d (see the comment **#3-3** for new Fig. 6).

(#3-10) • Minor re-writing at several places would clarify your conclusions more and support what your figures show. For example- Line 145 and 148: mention 30C after “grew normally with the compounds”, otherwise the fig does not match the sentence. Mention the temperature

after “an elongated morphology”. Also, the figures say 43C but the text says 42C. May be it’s a typo?

=> As the reviewer suggested, we changed the corresponding sentence to “grew normally at 30°C when treated with the compounds” (line 128/212). As for the part “died uniformly with an elongated morphology”, we previously reported that the elongated morphology is observed upon the overexpression of a hyperactive variant (DegP_R207P/Y444A) at both higher and lower temperatures (ref 16). Therefore, we changed it to “died uniformly with an elongated morphology at both higher and lower temperatures” (line 155-156/270-271). The mismatched temperatures are corrected to 42°C (see comment **#3-2** for new Fig. 4).

(#3-11) • Line 27: Normal temperature means optimal temperature or relatively lower temperature? Be specific. A little more background on why you are doing experiments at 30C not 37C or so would have been helpful.

=> The temperature in this sentence is further explained in the following clause, “at which DegP is not essential for cell viability”. Therefore, to avoid confusion, we deleted the word “normal” (line 24/30).

DegP is essential at neither 30°C nor 37°C, of which the latter is the optimal temperature for *e. coli* growth. Therefore, I think that doing the same experiments at 37°C is fine and would result in the same conclusion. However, it is known that the extracellular heat shock response (the σ^E pathway) is partially activated at 37°C, which means that there is a small level of misfolded protein stress. To avoid the misfolded proteins stress as much as possible while keeping the significant level of cell growth, we did experiments at 30°C, not 37°C.

(#3-12) • Line 105: From fig 2c and sup fig 4, the pentapeptide compound is permanently activating DegP. So stronger or permanent instead of just “good” may be?

=> As the reviewer suggested, we changed “good” to “permanent” (line 109/177).

(#3-13) • Line 140: The temperature dependent essentiality of DegP is an important factor featured in this work, so rewriting the sentence may make it more clear e.g. “temperature-dependent essential protease into a temperature independent toxic enzyme”.

=> As the reviewer suggested, we modified the sentence to “the tripodal compounds turn a temperature-dependent essential protease into a temperature-independent toxic enzyme” (line 121-123/205-207).

REVIEWERS' COMMENTS:

Reviewer #1 (Remarks to the Author):

In this work ("Over-activation of a Nonessential Bacterial Protease as an Antibiotic Strategy"), Cho and colleagues design small molecule that can cause allosteric activation of DegP and lead to growth inhibition due to this inappropriate activation. The authors further confirm the interaction between the compound and DegP by analyzing a co-crystal. The authors propose that antimicrobial over-activating non-essential pathways is an alternative to traditional antimicrobials inhibiting essential pathways.

Overall, this work would be interesting to the protease, envelope biogenesis, and antibiotic discovery fields and is generally scientifically sound. The first version of this manuscript raised several concerns that weakened the support for their model. In this new version of the manuscript, these concerns have been adequately addressed and their model is well supported. These changes have improved the paper.

Reviewer #2 (Remarks to the Author):

Most questions have been well addressed, except of question 2-1. As I pointed out that over-activation of DegP by the compound in cells should be verified, although this experiment cannot determine whether over-degradation of target OMPs will lead to cell lethality. Indeed, over-expression of DegP in E. coli cells will lead to a substantial decrease in the levels of abundant OMPs (OmpA, OmpF, OmpC). Accordingly, it is the same forth with chemical-induced over-activation of DegP in cels.

Reviewer #3 (Remarks to the Author):

The authors responded well to the previous review, and the paper is much improved. A few small writing changes are recommended.

There are a few run on sentences. e.g.: Line 20-23, 32-36,

Line 38 – use "provides" not provide.

Line 65 – what is the stoichiometry of the binding of Lpp to DegP ? It's not clear to my from this sentence why a tripodal compound should activate DegP. Do three Lpp molecules bind to one molecule of DegP? If so, please say so in the text, you only allude to it indirectly.

Line 98 – if you want people who aren't familiar with the entire DegP literature to be able to understand your paper, please explain what a cage means in this context. I literally have no clue. If you don't want to explain these things to a general audience, maybe consider a more specialist journal.

Line 165-166 – sorry, but this conclusion is still not warranted by this data. I feel confident from your other data that this is likely to be true, but the cell length data does not show this. There are a lot of ways to interfere with bacterial division, and you have no evidence that the degP overactive mutant and the compounds are doing the same thing to cell division, especially since the degree of effect on cell morphology is so different. Tone down this conclusion.

Reviewer #3 (Remarks to the Author):

The authors responded well to the previous review, and the paper is much improved. A few small writing changes are recommended.

(1) There are a few run on sentences. e.g.: Line 20-23, 32-36,

=> To improve readability, we modified these sentences.

As for line 20-23, we changed to “We demonstrated that tripodal peptidyl compounds that mimic DegP-activating lipoprotein variants that were previously reported to activate DegP allosterically activate DegP and inhibit the growth of an *Escherichia coli* strain...”.

The sentence in line 32-36 is modified to “Furthermore, there has been limited innovation in antibiotic development: most antibiotics target only four major cellular pathways—maintenance of membrane, and biosynthesis of nucleic acids, of proteins, and of cell wall peptidoglycan—and newly developed antibiotics are mostly derivatives of old scaffolds, which are more susceptible to existing resistance mechanisms, revealing the limited innovation in antibiotic development²⁻⁴.”

(2) Line 38 – use “provides” not provide.

=> It is corrected as suggested.

(3) Line 65 – what is the stoichiometry of the binding of Lpp to DegP ? It’s not clear to my from this sentence why a tripodal compound should activate DegP. Do three Lpp molecules bind to one molecule of DegP? If so, please say so in the text, you only allude to it indirectly.

=> We designed trimeric compounds because Lpp is a trimeric protein. The binding stoichiometry of Lpp and DegP does not matter at all at this stage. To clarify this point, we changed the sentence to “Because Lpp is a trimeric protein²³ and only the C-terminal region of the Lpp^{+Leu} trimer is critical for activity modulation^{16,22}, we reasoned that tripodal compounds that mimic this C-terminal region may function as DegP activators.”

(4) Line 98 – if you want people who aren’t familiar with the entire DegP literature to be able to understand your paper, please explain what a cage means in this context. I literally have no clue. If you don’t want to explain these things to a general audience, maybe consider a more specialist journal.

=> As the reviewer suggested, we added one sentence for the allosteric behaviors of DegP: “Previous report showed that substrate binding accompanies several allosteric behaviors of DegP: proteolytic activation, assembly of cage-like proteolytic chamber and positively cooperative binding of a substrate¹⁴. Aside from the allosteric activation, we also analyzed the effects of the two compounds on ~~the other allosteric behaviors of DegP~~, assembly of cages and cooperative binding of a substrate¹⁴, and found that both TMB_CYRKL and TMB_CYYKI assemble DegP cages, but that they differently affect the binding of model substrates (Supplementary Fig. 3).”

(5) Line 165-166 – sorry, but this conclusion is still not warranted by this data. I feel confident from your other data that this is likely to be true, but the cell length data does not show this.

There are a lot of ways to interfere with bacterial division, and you have no evidence that the degP overactive mutant and the compounds are doing the same thing to cell division, especially since the degree of effect on cell morphology is so different. Tone down this conclusion.

=> As the reviewer suggested, we further toned down the conclusion: “Although DegP^{R207P/Y444A} and TMB_CYYKI-activated DegP appear to differently affect cells~~—may degrade slightly different sets of proteins~~, these results suggest ~~are consistent with the idea that TMB_CYYKI-driven growth inhibition is not the result of~~ inhibits bacterial growth not by misfolded protein stress but by over-proteolysis.”